

# Density-dependent changes in the distribution of Southern Right Whales (*Eubalaena australis*) in the breeding ground Peninsula Valdés

Nicolas Sueyro[1,2], Enrique Alberto Crespo[1,2], Magdalena Arias[3] and Mariano Alberto Coscarella[1,2]

[1] Laboratorio de Mamíferos Marinos (LAMAMA), Centro para el Estudio de Sistemas Marinos (CESIMAR—CCT CENPAT-CONICET), Puerto Madryn, Chubut, Argentina
[2] Facultad de Ciencias Naturales y Ciencias de la Salud, Universidad Nacional de la Patagonia San Juan Bosco, Puerto Madryn, Chubut, Argentina
[3] Centro de Investigación Aplicada y Transferencia Tecnológica en Recursos Marinos Almirante Storni (CIMAS), San Antonio Oeste, Rio Negro, Argentina

## ABSTRACT

**Background:** The Southern Right Whale (*Eubalaena australis*) population of the South–western Atlantic Ocean is recovering. In the breeding ground of Peninsula Valdés, as a consequence of the population growth, expansion to new areas by some types of groups and a change in the habitat use patterns at the coastal area were recorded.

**Methods:** We analysed information gathered from aerial surveys conducted along the coast of Peninsula Valdés in 15 years of effective sampling in a 19-year span. These surveys were divided into four periods (1999–2000; 2004–2007; 2008–2012 and 2013–2016) and estimated the density of whales in a 620 km of coast divided into segments of five km.

**Results:** The density of the whales increased to near three whales per km² (averaged over each period) in the high-density areas. When this mean number was reached, the significant changes in density in the adjacent areas were detected in the following period. These changes were a decrease in density in the high-density areas and an increase of density in the low-density areas.

**Discussion:** We propose that a threshold in density elicits a response in habitat use, with the *Mother-calf* pairs remaining in the area, while the *other* groups are displaced to new areas.

Corresponding author
Nicolas Sueyro,
nicosueyro@cenpat-conicet.gob.ar

## INTRODUCTION

The Southern Right Whale (*Eubalaena australis*) has a circumpolar distribution in the Southern Hemisphere. This species was subjected to a commercial exploitation between the 19th and 20th centuries that put the species on the brink of extinction (*Richards, 2009*). The species was protected for the first time in 1936 and additionally, in 1986,

the moratorium on commercial catch established by the International Whaling Commission (IWC) came into force. By the mid-1970s several populations have shown evidence of recovery, with a doubling time of 10–12 years (*Bannister, 2001*; *Best, Brandão & Butterworth, 2001*; *Cooke, Rowntree & Payne, 2001*). However, there are other populations that are still very small and there is uncertainty about their recovery (*Galletti Vernazzani, Cabrera & Brownell, 2014*). Even today, the geographical distribution of all breeding and feeding areas of the species prior to its exploitation are unknown (*International Whaling Commission (IWC), 2001*).

The estimated population size for the species in 1997 was 7,500 animals (including 547 mature females, from Argentina and 659 from South Africa), with an estimated average growth rate for all populations in the Southern Hemisphere of 7.5% (*International Whaling Commission (IWC), 2012*). The breeding population of Peninsula Valdés is one of the best studied, with a long-term research programme carried out in the Peninsula Valdés area since 1970 (*Payne, 1986*). The population size and other parameters derived from capture–recapture models have been estimated, based on the individual recognition of the whales. The population growth rate was estimated to be around 8% by the early 1980s (*Payne, 1986*; *Payne et al., 1981*, *1990*; *Whitehead, Payne & Payne, 1986*), and recent studies estimate a 5.1% for 2010 (*Cooke, 2012*) and 6.5% for 2012 (*Cooke, Rowntree & Sironi, 2015*) using the same techniques.

The process of gathering the information from photo-identification is time-consuming, and there was a need for a quick assessment of the number of whales and population trend by the authorities. In 1999 a protocol involving coastal aerial survey with the direct counting of marine mammals in the Peninsula Valdés area was started (*Crespo et al., in press*). *Crespo et al. (in press)* estimated for the year 2007 an increasing rate of 6.22%. For the year 2014, the models indicated that the population grew at a rate of 3.23% per year. However, the reduction in the rate of increase was not uniform across the different groups that comprise the population. The estimate of the growth rate for offspring born in the Peninsula Valdés was 5.54% per year between 1999 and 2014 (*Crespo et al., in press*), almost doubling the rate of increase for the whole population during the same period. The rate of increase of the other class groups (namely *Solitary Individuals* and *Breeding groups*) is now close to 0% (*Crespo et al., in press*). During the first sampling seasons, all the types of groups were close to the coast. As the population grew the *Mother-calf* pairs remained close to the shore, while the other types of groups were displaced farther off from it. This change in Peninsula Valdés distribution could mean that the coastal zone (within two km from shore) is close to it carrying capacity (*Crespo et al., in press*). Therefore, in spite of the overall decrease in the population increasing rate observed in the sampling area, it is proposed that the southern right whale stock of the south–western Atlantic is still growing at a rate that would be the combination of its growth in the Peninsula Valdés area and the occupation rate of other zones (*Crespo et al., in press*). If this scenario is correct, whales should begin to move to other less dense regions (i.e. suboptimal habitats) where the growth rate should be higher (*Hobbs & Hanley, 1990*; *Verner, Morrison & Ralph, 1986*). There are different indicators supporting this hypothesis for Peninsula Valdés as evidenced by the number of whales occupying areas with deeper

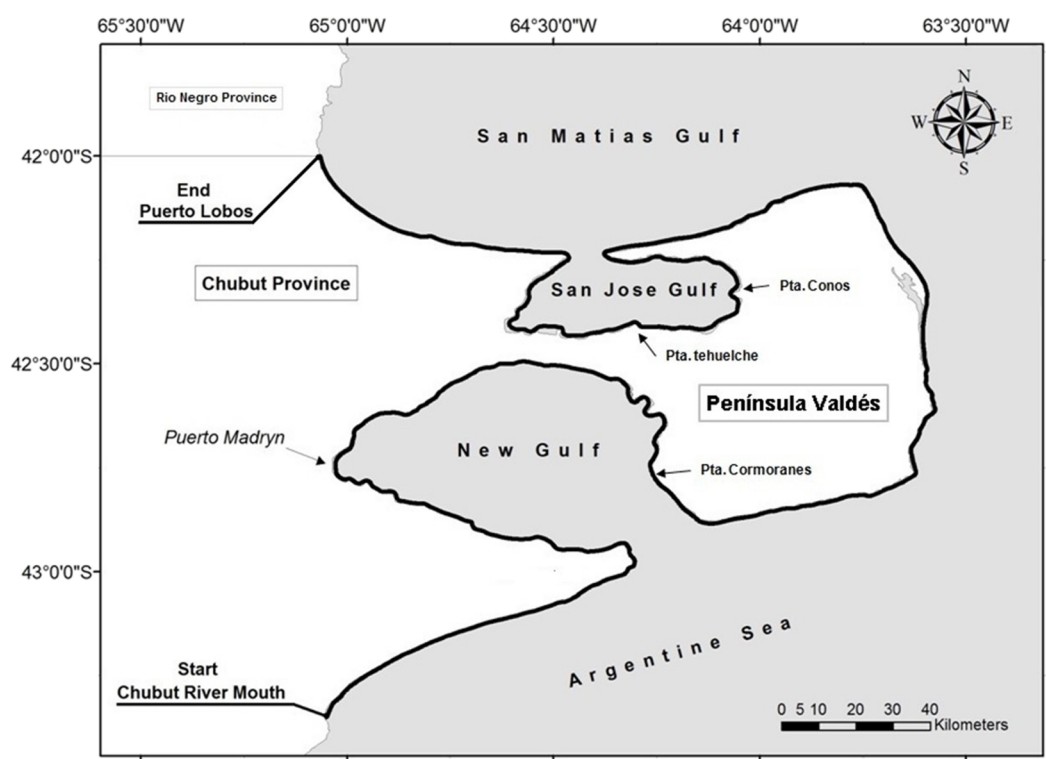

**Figure 1 Sampling area.** The thick black line along the coast represent the surveyed area.

waters in Peninsula Valdés and the growing number of whales observed in Golfo San Matías, Buenos Aires, Uruguay and Santa Catarina in southern Brazil (*Crespo et al., in press*; *Groch et al., 2005*; *International Whaling Commission (IWC), 2011*).

The expansion of whales to other areas may be determined by the priority occupation of mothers with offspring in optimal habitat areas (*Barendse & Best, 2014*; *Carroll et al., 2014*; *Danilewicz et al., 2016*). If this is the case, there should be a threshold density in which the *Other* groups increase their density in adjacent areas (e.g. nonpreferred by *Mother-calf* pairs). Similar trends have also been recorded in other southern right whale stocks from South Africa, New Zealand and southern Brazil, where *Solitary Individuals* and *Breeding* Groups move to new sites, outside the established breeding area (*Barendse & Best, 2014*; *Carroll et al., 2014*; *Danilewicz et al., 2016*).

## METHODS

The censuses were carried out from a single-engine high-wing CESSNA B-182 aircraft, flying at a constant height of 500 feet (152 m) and at 80/90 knots every 45 days from April to December each year (*Crespo et al., in press*). In each survey a distance of 620 km was covered in 5 h of flight, flying from south to north along the coast. The surveyed area (Fig. 1) is located between the mouth of the Rio Chubut and Puerto Lobos on the border with the province of Rio Negro. The width of the strip is composed by 500 m from the coast plus approximately 1,000 m from the plane to the open sea, composing a

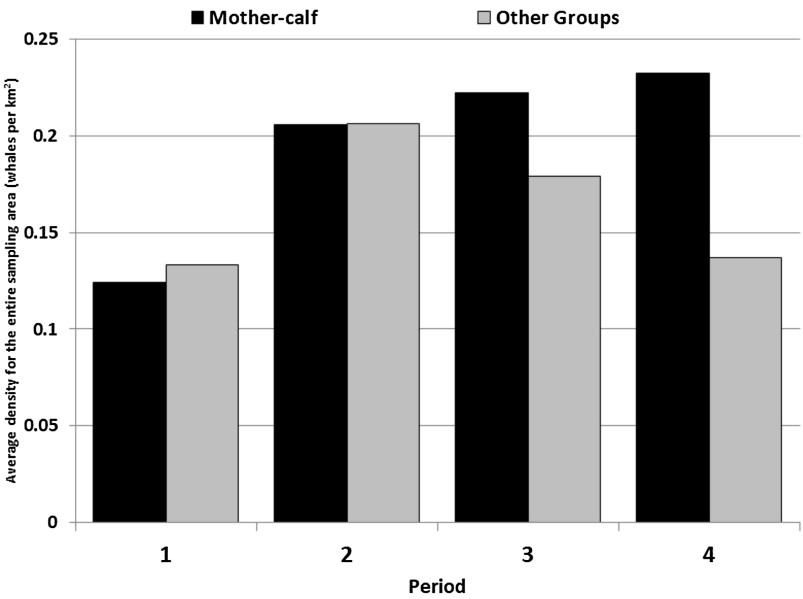

**Figure 2 Variation of the average density of the sampling area for each established period.**

surveyed strip of 1,500 m (*Crespo et al., in press*). This strip is set to cover the 'whale-road' as described by *Payne (1986)*, where more than 90% of the whales in the area concentrate near the coast in shallow waters.

The team comprised a pilot, a recorder sitting next to the pilot and two observers in the rear seats, one on the left and one on the right side of the aircraft (*Crespo et al., in press*). The observations were made with the naked eye, and the information was recorded in spreadsheets or tablet applications developed ad-hoc with Cyber Tracker™. Information on the group composition was recorded, including *Mother-calf* pairs, *Solitary Individuals* or *Breeding* groups comprising a female and several males (*Crespo et al., in press*). For the analysis in this work *Solitary Individuals* and *Breeding* groups were pooled in *Other* groups, as a group category opposite to the *Mother-calf* pairs. Along with the type of group we recorded the position registered with a handheld GPS, the number of individuals and the sea state on the Beaufort scale. Flights were suspended when the visibility conditions were not optimal, either because of fog or the sea state exceeded the level 3 on the Beaufort scale. Data were entered into a database developed in Access-Microsoft® specifically for this purpose. Fields in the database included time and position, number of animals, side of the sighting, type of group and other ancillary information.

The information was grouped into four periods, making each period as similar as possible considering the number of flights. A total of 58 aerial surveys were clumped in four periods: period 1 from 1999 to 2000 (eight flights) and the other three periods comprised from 2004 to 2007 for period 2 (18 flights), period 3 from 2008 to 2012 (17 flights) and period 4 from 2013 to 2016 (15 flights). No flights were performed during the period 2001–2003.

The surveyed coast was divided into segments of five km in length, totalling 124 segments for the 620 km sampled in each flight, where the zero km is the mouth of the Chubut River (Fig. 2). The densities were calculated by dividing the number of whales

counted in each segment, weighted by the number of flights performed per period and the area of each segment, calculated as the five km segment by 1.5 km (bandwidth), and thus each segment accounted for a 7.5 km$^2$. The length of the segment was chosen following *Rowntree, Payne & Schell (2001)*, who divided the coast into five km segments to evaluate the distribution of the Southern Right Whales.

Preliminary examination of data led us to define two high densities zones, one inside Golfo Nuevo from Puerto Madryn to Punta Cormoranes and other inside Golfo San José, from Punta Conos to Punta Tehuelche (Fig. 3). These areas coincide with the ones defined previously by *Rowntree, Payne & Schell (2001)* for the areas selected by whales in the early 1990s. We also defined the low-density zones as those outside the high-density zones (i.e. Chubut River mouth-Puerto Madryn; Punta Cormoranes-Punta Conos; Punta Tehuelche-Puerto Lobos). Differences in densities among the periods in these zones were assessed by Mann–Whitney $U$-tests (*Zar, 2010*).

The permit was granted by the Secretaría de Turismo y Áreas Protegidas of the Province of Chubut (issued for the last time under permit number 93-SsCyAP/15).

## RESULTS

The overall density of both group categories for the whole surveyed area shows for the first period a similar density for *Mother-calf* pairs and *Other* groups. For the second period, the observed increase of density is similar for both categories (Fig. 2). During the third and the fourth period, the density of *Mother-calf* pairs slightly increases, while the *Other* group's density decreases (Fig. 2).

The coastal area defined as high densities zones are located between the segments 24 (Puerto Madryn) and 49 (Punta Cormoranes) (*high-density 1*) within the Golfo Nuevo and between the segments 90 (Punta Conos) and 101 (Punta Tehuelche) (*high-density 2*) within the Golfo San José (Fig. 3). The highest densities in any of the four periods were observed at the Doradillo (segment 28 and 29) and Playa Fracaso (segment 97 and 98), reaching a maximum estimated of 3.15 whales per km$^2$ during the first period for all type of groups.

Figure 4 shows the density changes in the two areas of high density of animals. In the *high-density 1*, the *Mother-calf* pairs increased their density in the second and third period with respect to the first ($U_{1,2}$: 181.5/$U_{1,3}$: 157.5; $p < 0.05$) but a great variation was observed in the fourth period. The density of the *Other* groups category increases, but with a subsequent decrease ($U_{1,2}$: 179/$U_{2,4}$: 426; $p < 0.05$).

In the *high-density 2*, a similar pattern is observed, an increase in the density of *Mother-calf* pairs ($U_{1,3}$: 41.5; $p < 0.05$) that is sustained in time. For the groups without calves or the *Other* groups, the initial increase is followed by a marked decrease in the last period ($U_{2,4}$: 143/$U_{3,4}$: 147.5; $p < 0.05$).

Figure 5 shows the changes in the low-density zones. *Mother-calf* pairs in the *low-density 1* increased their density during the third period ($U_{2,3}$: 215; $p < 0.05$). Also, an increase in the density of the *Other* groups from the first to the second period can be observed ($U_{1,2}$: 205; $p < 0.05$). Afterwards a decrease in the density from the second to the third ($U_{2,3}$: 540; $p < 0.05$) was recorded to the groups without calves. In the *low-density*

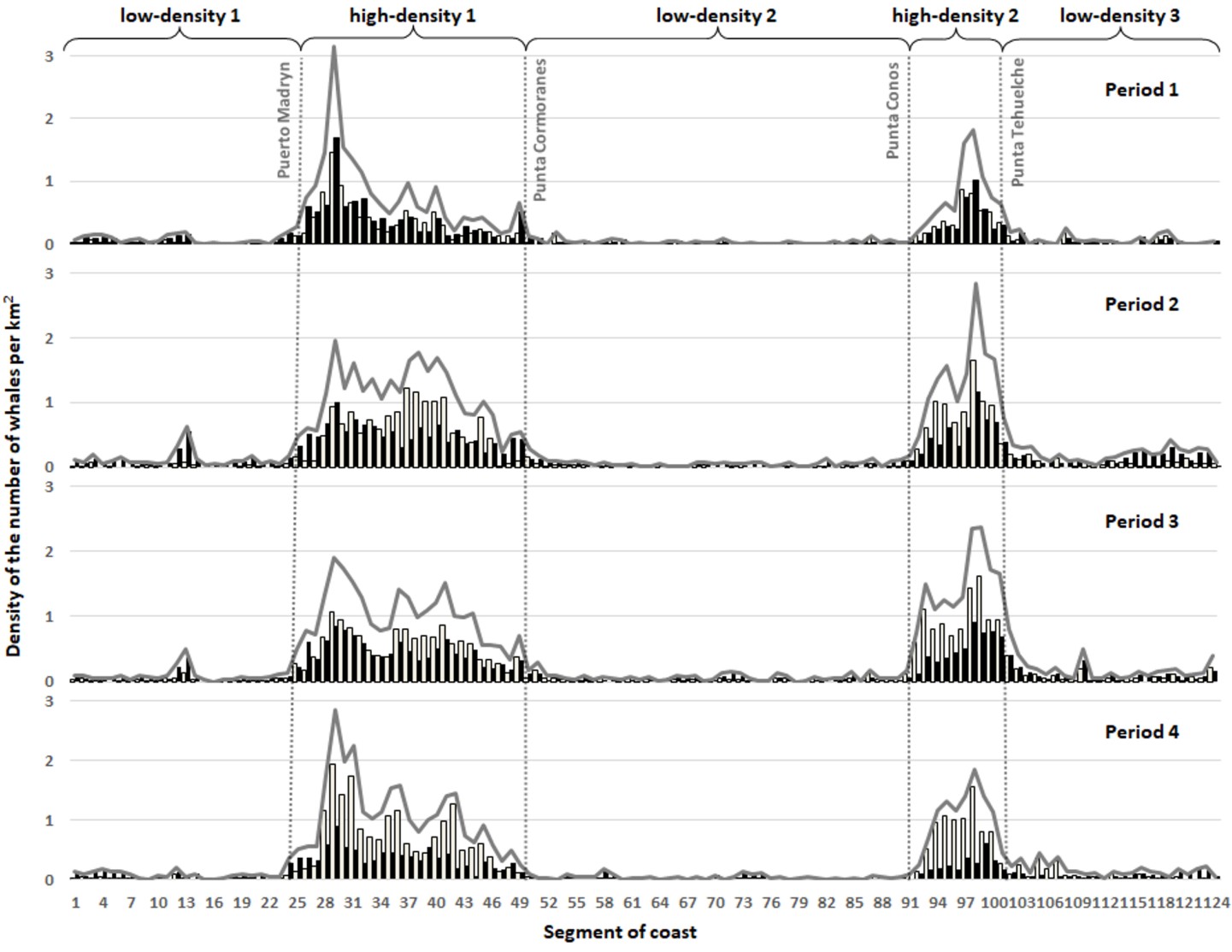

**Figure 3 Density of the number of whales per km² in each segment of five km performed in the four established periods for all the counted whales in the surveyed area.** *Mother-calf* pairs: white bars, *Other* groups: Black bar, All groups: grey continuous line.

2 the *Mother-calf* pairs increased their density only during the third period ($U_{1,3}$: 489/$U_{2,3}$: 437/$U_{3,4}$: 1,054; $p < 0.05$) decreasing afterwards. For the *Other* groups there was an increase in the second period and it remained higher than the first period later ($U_{1,2}$: 372/$U_{1,3}$: 519.5/$U_{1,4}$: 511.5/$U_{2,4}$: 994.5; $p < 0.05$). In the *low-density 3* for the density of *Mother-calf* pairs increases during the whole period ($U_{1,2}$: 119/$U_{1,3}$: 109/$U_{1,4}$: 116; $p < 0.05$). The *Other* groups show an increase in density in the second period and a sharp decrease later on ($U_{1,2}$: 56/$U_{1,3}$: 82/$U_{1,4}$: 71/$U_{2,4}$: 308; $p < 0.05$).

Overall, when in a particular period in a high-density area the mean density for the period reaches around three whales per squared km², the next period is characterised by

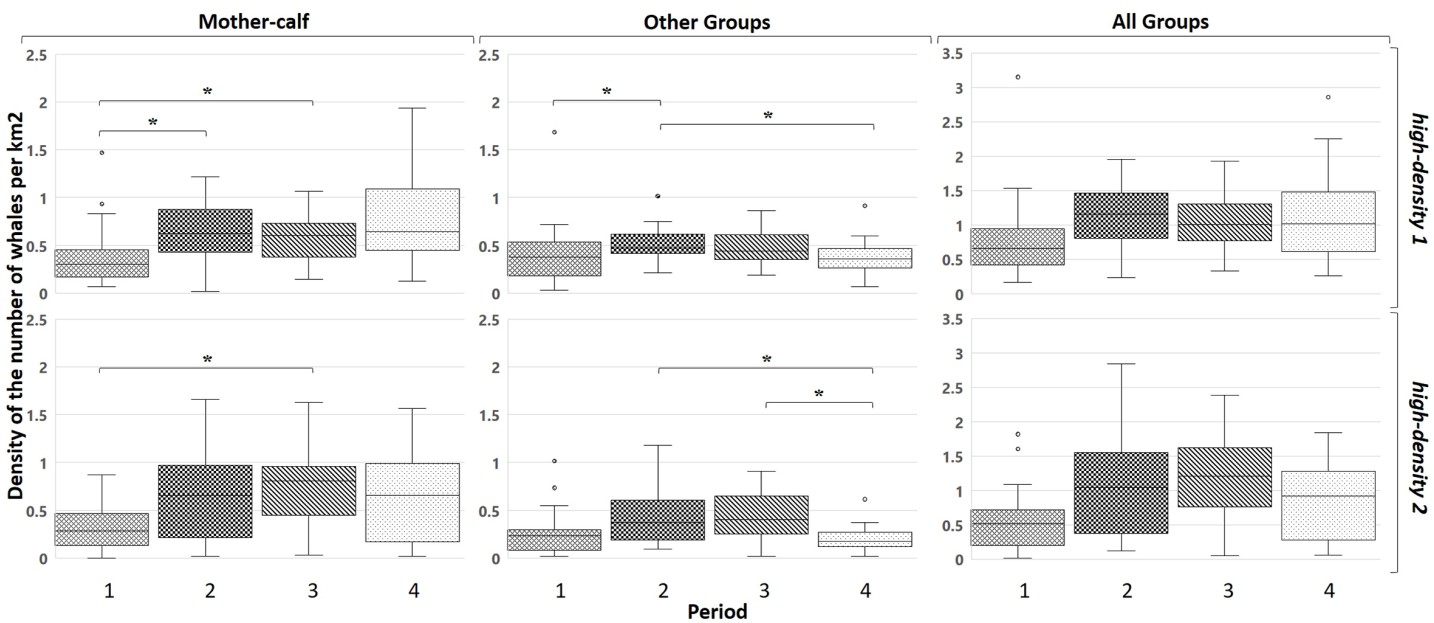

**Figure 4 Variation of whale densities in the four periods for the three groups classified in the two high-density zones.** The brackets with the asterisk show the periods where there is a significant difference.

**Figure 5 Mean densities of whales in the four periods for the three groups classified in the three low-density zones.** The brackets with the asterisk show the periods where there is a significant difference.

a decrease of density in it. The process is also accompanied by an increase of density in the nearby low-density areas.

## DISCUSSION

This is the first study that proposes that a threshold in whale's density in breeding areas triggers a density-dependent response, it has been observed in other species of smaller size and in its majority of terrestrial environments (*Matthysen, 2005*). This response includes the movement of *Solitary Individuals* and *Breeding* Groups (*Other* Groups) to adjacent areas when the average density in the area is close to three whales per km$^2$. This must be taken with caution since it is an average from April to December for each period (*Crespo et al., in press*). In any given season, as much as 15.87 whales per km$^2$ can be found in the El Doradillo area for a few days (up to 15 days approximately) from the end of July until October. The interannual variation in the exact moment the maximum of whales is present in the area precludes us to use the census near the peak of the season because the surveys cannot always coincide with this peak in the number of whales. Hence, the density-dependent process proposed here cannot be observed using only the peak of the season but using the average of the density. Changes in the distribution of the Southern Right Whales in Peninsula Valdés were reported by *Rowntree, Payne & Schell (2001)* during the late 1980s. *Rowntree, Payne & Schell (2001)* recorded a movement of whale breeding areas from the outer coast of Peninsula Valdés into the gulfs (Golfo Nuevo and Golfo San José), but no mechanism was proposed. The areas reported as new for the 1990s in Peninsula Valdés are the same as those observed in this work for the first period (Fig. 2). *Rowntree, Payne & Schell (2001)* considered several factors such as gull inflicted wounds or undetected changes in the environment and topography of the area as the possible causation of the observed shift in distribution. Our data lead us to propose that density changes (and probably related social causes) can be the main mechanism that promotes the search for new areas and the expansion of the occupied coast in Peninsula Valdés breeding ground. This mechanism might not have been the same mechanism that drove the changes observed during the late 1980s. Other social interactions not previously described in the area are being recorded. *Mother-calf* interactions and mating groups were almost exclusively the only social interaction recorded, but nowadays is possible to observe adult individuals without calves engaged in the same activities (e.g. travelling together or socialize without forming a *Breeding* group) (*Arias et al., 2017*). Also, cooperative feeding was recorded recently (*Argüelles, 2017*), and hence these new interactions never reported before may be shaping the social behaviour of whales in the area. These social aspects of the whales may have not been previously reported due to the low number of individuals until recent years.

*Mother-calf* pairs continue to select areas used in the late 1990s, the so-called high-density zones (Fig. 4). The density in these areas increased, but differentially by type of groups: while *Mother-calf* increased their density, the *Other* groups continue to select this area until an average threshold of three whales per km$^2$ is reached. In every considered period, this is mainly due to the increase in density of *Mother-calf*. The only fraction of the population that is still growing, since the other type of groups have not

shown positive increasing rates in the last few years (*Crespo et al., in press*). During the first period, the average annual density reached near three whales per km$^2$ in the area Puerto Madryn- El Doradillo. In the following period a change in the density of whales occurred (Fig. 4); not only was there an increase in the *Mother-calf* pairs density, but the *Other* groups are more prone to be found in peripheral areas of less density (e.g. *low-density 1*), as shown in Fig. 3. The same pattern can be found in Golfo San José high-density area during the second period. After the mean density approached to three whales per km$^2$, the density of the *Other* groups increased in the *low-density 3* (Fig. 5).

In the fourth period, it is observed that the average annual density of whales in the Doradillo area is close to three whales per km$^2$. If the same pattern is repeated in the next period, we could hypothesize that an expansion of part of the whales to areas with physical and biological conditions similar to those found in Peninsula Valdés will occur in the next few years, as well as a new increase in the low-density areas.

In a context of population growth, the expansion into new areas is been driven by *Solitary Individuals* and *Breeding* groups (*Arias et al., 2018*). The optimum areas are first occupied by the *Mother-calf* pairs, as the density of this type group increases, the rest of the groups are displaced to suboptimal zones, as observed in the San Matias Gulf where the first groups in the area were the *Solitary Individuals* and *Breeding* groups (*Svendsen, 2013*).

This kind of mechanism was recorded in other mammals. When the red deer (*Cervus elaphus*) population of the island of Rum doubled its size, females presented on average a greater spatial distance among them (*Albon et al., 1992*). In the case of roe deer (*Cupreolus capreolus*) their increase in density caused a change in the habitat use of young males and later of adult males. While females continue to use the habitat used by the population in the past (i.e. optimal habitat), younger males tended to move to other areas (*Vincent et al., 1995*). Our results indicate that the recolonization process started at least in the mid-2000s when whales changed both the way they use the habitat related to the type of groups and the areas where they could be found. The expansion of these groups to other areas was observed in the province of Rio Negro with the presence of whales in the area near the San Antonio Bay. In this area, more than 80% of whales are *Solitary Individuals* and *Breeding* Groups (*Arias et al., 2017*; *Crespo et al., in press*). These movements of *Solitary Individuals* and *Breeding* Groups move to new sites, outside the established breeding area has also been recorded in other southern right whale stocks from South Africa, New Zealand and southern Brazil, but no mechanism was ever proposed (*Barendse & Best, 2014*; *Carroll et al., 2014*; *Danilewicz et al., 2016*).

## CONCLUSION

The growth and expansion of the southern right whale population in Peninsula Valdés are proposed to being modelled by density-dependent determinants. One proposed mechanism is related to social mediated factors. It is important continuing to monitor the rate of increase in the core areas as well as the densities in these areas. The rate of increase of the population is now the combination of the increase recorded in Peninsula Valdés and the growth experienced while probably recolonizing ancient habitat (*Arias*

*et al., 2018*; *Crespo et al., in press*). Also, it is important to evaluate the habitat suitability of different areas, and to test if the mean density of around three individuals per km$^2$ is an actual threshold that is also found outside Peninsula Valdés; and if so, which are the social causes that trigger this density-dependent response.

## ACKNOWLEDGEMENTS

This paper is dedicated to our colleague Susana N. Pedraza; she was the most influential quantitative researcher at the Marine Mammal Laboratory. We thank S. Dans, G. Svendsen, M. Degrati, F. Grandi, G. Garaffo, B. Berón Vera, L. Hardtke, F. García, A. Carribero, C. Giesse, D. Valés, R. Loizaga, J. Klaich, S. Leonardi, N. Martínez, V. Milano, M. Arias and C. Durante helped as observers or recorders in the flights. We thank several pilots throughout the period but mainly Peter Dominguez.

### Funding

This project was supported along the years by Fundación Vida Silvestre Argentina (thanks to Javier Corcuera and Manolo Arias), BIOCON 04 BBVA Foundation, CONICET, ANPCyT, National University of Patagonia, GEF/PNUD 02/018, Southern Spirit SRL (thanks to Tiño Resnik) and Secretaría de Turismo de la Provincia de Chubut. Logistical support was provided by CENPAT and National University of Patagonia. The funders had no role in study design, data collection and analysis, decision to publish, or preparation of the manuscript.

### Grant Disclosures

The following grant information was disclosed by the authors:
Fundación Vida Silvestre Argentina (thanks to Javier Corcuera and Manolo Arias).
BIOCON 04 BBVA Foundation, CONICET, ANPCyT, National University of Patagonia, GEF/PNUD 02/018.
Southern Spirit SRL (thanks to Tiño Resnik) and Secretaría de Turismo de la Provincia de Chubut.
CENPAT and National University of Patagonia.

### Competing Interests

The authors declare that they have no competing interests.

### Author Contributions

- Nicolas Sueyro conceived and designed the experiments, performed the experiments, analyzed the data, contributed reagents/materials/analysis tools, prepared figures and/or tables, authored or reviewed drafts of the paper, approved the final draft.
- Enrique Alberto Crespo conceived and designed the experiments, contributed reagents/materials/analysis tools, authored or reviewed drafts of the paper, approved the final draft, funding for the aerial censuses and began with the samplings in 1999.

- Magdalena Arias conceived and designed the experiments, performed the experiments, authored or reviewed drafts of the paper, approved the final draft.
- Mariano Alberto Coscarella conceived and designed the experiments, performed the experiments, contributed reagents/materials/analysis tools, authored or reviewed drafts of the paper, approved the final draft.

## Field Study Permissions

The following information was supplied relating to field study approvals (i.e. approving body and any reference numbers):

The permit was granted by the Secretaría de Turismo y Áreas Protegidas of the Province of Chubut, (issued for the last time under permit number 93-SsCyAP/15).

## Data Availablity

The raw data is provided in the Supplemental Files.

## Supplemental Information

Supplemental information for this article can be found online at http://dx.doi.org/10.7717/peerj.5957#supplemental-information.

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
