# Peer review of "Density-dependent changes in the distribution of Southern Right Whales (Eubalaena australis) in the breeding ground Peninsula Valdés"

_PeerJ, doi:10.7717/peerj.5957_

## Round 0.1 · original submission · Major Revisions

The paper present data on density-dependent changes in the distribution on SRW in the breeding ground of Patagonia, providing very good data on this population. Both reviewers agree you have to increase paper clarity. Main corrections as, to make clear for the reader if the areas of higher and lower density were identified prior or during data analysis, to separate speculation and conclusions providing their support, and some suggestions that for sure will make the Ms clearer (to redefine categories of groups being compared) and minor editorial correctios as to be more consistent thorough the Ms with the use of commas vs periods, etc. I am also sending by e mail a tracked correction made for one of the reviewers. Please follow their suggestions

·

Basic reporting

This paper is reporting on the density-dependent changes in the distribution of southern right whales in the breeding round Peninsula Valdes. The authors provide good data on a population of whales that needs more information, particularly as things are changing – this kind of monitoring is vital. The overall idea of the paper comes across, but there needs to be more work done on the structure and flow of the paper to increase clarity and readability. The basic reporting here is good, but needs improvement for publication. There is some inconsistency with numbers (using commas vs. periods), and quite a few grammatical/spelling errors. Some of the sentences are confusing and need to be reworded for clarity and understanding. The intro and background do give context, however I think much can be improved here. Improvements can be made to sentence/paragraph structure, clarity of details and providing more detail/context in the introduction, which can then be discussed further in the discussion. In particular the authors should make a clear case as to why their study needed to be done, as much of the references are recent, and seem to cover much of the same information they are showing. I believe there is some overlap, but that this study was needed to explicitly (and numerically) show the changes occurring, but the authors need to make sure they frame this correctly throughout the paper in relation to all the other studies done on this population. See general comments for specific details.
Figures are relevant and show the data clearly, but need improvement on some captions and data labels.

Experimental design

This paper seems in the scope of the PeerJ journal. Overall the methods are good, but more details/increased clarity is needed in some spots. The research question is defined, however I think it can be made clearer (in relation to what was stated above in the basic reporting section) – in particular how does it fill an identified knowledge gap with all the other recent research being referenced. See general comments for specific details.

Validity of the findings

The authors provide valid results related to the question being asked, which is a valid question to investigate. Overall the conclusions are well stated and link to the original research question, however restructuring is needed in the discussion to more clearly define speculation/conclusions and providing support for each. I cannot say whether the statistics are completely appropriate or not. See general comments for specific details.

Additional comments

Abstract
Line 17: Southern Right Whales are (not is)
Line 18-21: combine this two sentences – there is no need to say, ‘as a consequence of population increase’ twice, since it is the same reason for both results. (so …. were recorded, and distribution and density in the core area are suspected.
Line 23: Please state what the 4 periods were.
Line 25: capitalize the. Can you give a little more results here – there are some interesting things you bring up in the discussion part, but you can’t get any of that from this short results section.
Introduction
Line 31: object or subject?
Line 31: I would use regular numbers here, unless this is a formatting issue for the journal. Not everyone can read roman numerals this large.
Line 46: There are a lot of years being thrown around here – when is 8% at that time? The 80s?
Line 47-53: It is a little confusing how the Cooke and Crespo studies are related (or not). They seem to overlap in time, but are showing the same thing? Are all the numbers population increases/decreases – and if so, then having the sentence line 49-50 in the middle is confusing because it makes it seem like the next sentences about the Crespo study are about that maximum number of whales in Aug/Sept vs. the population increases like the Cooke studies. Please rework this section to make it more clear chronologically and in what each study was showing relating to the point you are trying to make (which I think is that the population increases were decreasing in more recent years).
Line 57: Please add a sentence in here describing the types of all class groups involved.
Line 59-60: This sentence is not clear – where are the solitary and breeding groups?
Line 63: add population before increase – so it is a decrease in the rate of population increase (makes this more clear).
Line 68: supporting
Line 70: remove the (so in sourthern Brazil).

Methods
Line 89: remove on of the ‘on the’
Line 91: Please explain more the Breeding groups category – why n-1 males?
Line 95: on the Beaufort scale.
Line 96: What database? Please describe your custom database (and what software it runs on, i.e. is a access based, etc?)
Line 99: This is the info that would be helpful in the abstract (similar number of flights being the reason for clumping).
Line 104: Remove ‘later the coast of’ Start with The surveyed coast was divided….
Line 104-111: Is this density different than the one in line 102-103? Please clarify. Also, it is unclear what your calculation is; you divided the number of whales be what? You just explain the weighted part – is that what you divided it by? Perhaps an example calculation would be helpful here.
Line 112: This seems like results, not methods – didn’t the data show you what was high vs. low density? If not, then please explain more here what these are and how they were determined. I think perhaps you did get this from the data, if so then please explain this here. I.e. based on the data we determined high density to be xxx (or more) and low density areas to be xxx (or less) and used Mann-Whitney U tests to determine if the differences between these areas were statistically significant.

Results
Line 131-132: Put a periods after (segment 97 and 98), the rest of the sentence is redundant since you stated this in the beginning.
Line 132: Where and what time was this maximum mean density found?
Line 148: For the Other groups there was (not the was)
Line 151: only (not obly)
Line 126-154: For those not familiar with the area the results section is not clear, it is hard to follow with all the names and segment numbers. I would suggest subtitling two sections under the Results, 1. High density areas, 2. Low density areas – that will make which overall areas you are talking about. I would suggest that naming them isn’t needed in the text here, perhaps giving them numbers on the figure would help, then you can just put the numbers here. I would also like to see a clearer layout of how many high vs. low density zones there are, and how many showed increase or decrease (i.e. did all high density areas show the same trend?). Although I get the idea here, I think this section can be reworked to be clearer and have more information included.

Discussion
Line 159: This is an interesting idea, and the fact that you have a number for the threshold – but that wasn’t made clear in the results. Was it at that number when you started seeing shifts in the m/c vs other groups increase/decreasing in particular areas? If so, lay that out in the results more clearly, then discuss it here.
Line 161 – I think it should be 15.87 (not 15,87)
Line 162: description of what processes? This sentence is confusing, and what average are you using, the 15.87 or the 3?
Line 163: These kinds of changes, also southern right whales (not whale). This sentence is also confusing – what kinds of changes? The decreases in other groups you are documenting? Or the increases seen in Aug/Sept?
Also in the rest of the paper Southern Right Whale is capitalized, but here and afterwards it is not. Please be consistent throughout the paper.
Line 165: Rowntree et al. proposed (not propose).
Line 168: several factors such as…
Line 170: I would hesitate linking density to socially so directly. Although they can be linked, they aren’t always, as you could have density effects in relation to foraging that would not necessarily have a social component (there is just how much food there is to go around, regardless of how the animals may interact). I think that there very well may be a social component, but the data related here cannot state a clear link, so I would caution the authors to suggest this, but not state as fact.
Line 175-76: the sentence is not clear – if they are the same activities, how are they new interactions? Please state what the new interactions are, and who is doing them more clearly. Then this is where you can suggest a little more strongly that social causes may be a large part of the distribution shift, since there is other evidence besides just the density shifts you show. *also after getting to the end I see you bring more evidence… I suggest you put this statement toward the end, after you have shown other support. So start here with suggesting this might be a cause, then go through and discuss why based on your results and other studies in the literature.
Line 181: This is the info you want in the results, so you can reference again here.
Line 185: not only was there….
Line 187: remove ‘in the second period’ – this is redundant since you started the beginning of the sentence talking about the next period, after the first.
Line 199: groups move (remove are a). Also this sentence is a good one, and may be useful in the introduction supporting the need for the current study to document the actual density changes, and see if they follow the same trend in this area. I would perhaps also note in the text where the Svendsen study was done – to show that if it was done in another area, that now your study will investigate this trend in your study area; and if it was done in the same area, you should explain why your study was needed, as it sounds like similar information was presented in that study).
Line 200: mechanism was
Line 202-204: How is this different than the red deer you just talked about? You don’t talk about abandonment of areas in any species, there is no need for this sentence in relation to fission/fusion societies in particular. I would remove it, but if you want to keep it then please state more clearly other examples of different populations that increase by abandonment, along with others that support your idea about fission/fusion being different.
I would also move the roe deer example up after the red deer… put them together since your mechanism is both increasing spatial distance and the changes in habitat use that differ by group type. Then show how that happened in your study, and in other studies you have mentioned. This will give a better flow to the paragraph and a clearer discussion.
Line 205: remove of (so , new matrilineal lines). Also change to ‘establish new breeding areas in time’.
Line 206: remove the from the roe deer – it makes is sound like you have talked about the roe deer before when you have not.
Line 210 remove the comma
Line 211: the groups used the habitat differently? That is an interesting point that should be discussed more and probably brought up in the introduction, again as support for why you are doing your study.
Line 215- 216: again info that would be good as background in the introduction.

Conclusion
Line 221: One proposed mechanism is…
Line 223: well
Line 225: What is ancient habitat? I thought the new areas were new… are they moving into new areas or recolonizing older areas? This should be stated more clearly in the introduction.

Figures
Figure captions should all have punctuation.
Figure 1: is it 0.05 or 0,05? Be consistent in the manuscript. Also what is the unit for density here – whale/km or whale/km2? Please state it clearly in the graph. I would also highlight the high and low density zones here if possible (or at the very least the 2 high density zones).
Figure 3: I would remove the last part of the sentence ‘without discriminating the group type’ – you did discriminate the group type in the graph, and have them all combined.
Figure 4 and 5: the should be capitalized.

·

Basic reporting

The introduction section is well written and provides a good overview of the population being studied. As the article brings a new concept which has never been demonstrated for baleen whales in breeding areas, I believe that an overview of the topic at the end of the introduction section is crucial. The authors make reference to “Matthysen E. 2005. Density-dependent dispersal in birds and mammals. Ecography 28:403” but very superficially.


Along the text (see the pdf and the word files) I have made suggestions to improve the clarity and the grammar. I believe that a table would be a good support to summarise the periods and the data being used for the reader. I provided few comments to the figure 1, in which the areas of higher density could be illustrated to guide the readers.

Experimental design

-I suggest the adoption of another term for the categories of groups being compared: Mother-calf pairs and groups without calves, which includes solitary and breeding groups. What about dyads or trios? Are these group categories observed in the area?
-In the methods section, it is not clear for the reader if the areas of higher and lower density were defined prior or identified during data analysis.
- The average of mother-calf pairs never reaches 3 whales per km2. The density of all groups combined is greater than 3 in the first period in the segments that correspond to El Doradillo area. In the discussion section the authors state: “This response includes the movement of Solitary Individuals and Breeding Groups (Other Groups) to adjacent areas when the average density in the area is close to 3 whales per km2.” and: “During the peak of the season, as much as 15,87 whales per km2 can be found in the El Doradillo area but the description of the process can be better viewed using the average.”
Why is 3 whales per km the threshold? It is really not clear how the threshold was established.

-Was it the aim of the article to establish the threshold? If yes, the methods to define it have to be better described. If the aim was only to describe the probable process that is guiding habitat use patterns, is the value really necessary? If at the peak of the season density can be five times higher in high-density areas, is the number that matters, or the process? If it is the number, again, the method to identify it is not clear. I think that it is extremely important here that the assumptions made during the data analysis be clearly stated. You have pooled all observations together to get a density for a period. Why? Why haven't you calculated the density by month or period within a year and after made an average to take your n into consideration (number of flights). These points must be clear in your methods and discussed. If your goal was to characterise the spatial process, then, the discussion (mainly) must be reviewed in consequence.

Validity of the findings

The article uses a large data set to demonstrate an ongoing process with important consequences on the species habitat use patterns. To my knowledge, a density-dependent process has never been demonstrated for baleen whales in breeding areas. The methods need to be better presented and the results better discussed in order to allow the reader to replicate the analysis and to validate the analysis that was made.

The authors make often reference to the article of Crespo et al which is in press. I assume that the same data set is used in both and different aspects are presented in each. However, in different points of the discussion of the present article, it is not clear if the concept of density-dependence is being proposed here, or by both. The same argument stands for the article of Arias et al.

Additional comments

Check the spelling of all locations. Doradillo or El Doradillo?
I have made all my comments along the text. I started by the pdf and moved to the word file, I am sorry for that!

---

## Round 0.2 · accepted · Accept

Many thanks for making the corrections suggested by both Reviewers.
I look forward to seeing the paper published.

#